# Novel Auger-Electron-Emitting ^191^Pt-Labeled Pyrrole–Imidazole Polyamide Targeting MYCN Increases Cytotoxicity and Cytosolic dsDNA Granules in MYCN-Amplified Neuroblastoma

**DOI:** 10.3390/ph16111526

**Published:** 2023-10-27

**Authors:** Honoka Obata, Atsushi B. Tsuji, Hitomi Sudo, Aya Sugyo, Kaori Hashiya, Hayato Ikeda, Masatoshi Itoh, Katsuyuki Minegishi, Kotaro Nagatsu, Mikako Ogawa, Toshikazu Bando, Hiroshi Sugiyama, Ming-Rong Zhang

**Affiliations:** 1Department of Advanced Nuclear Medicine Sciences, National Institutes for Quantum Science and Technology (QST), 4-9-1 Anagawa, Inage-ku, Chiba 263-8555, Japan; obata.honoka@qst.go.jp (H.O.);; 2Department of Molecular Imaging and Theranostics, National Institutes for Quantum Science and Technology (QST), 4-9-1 Anagawa, Inage-ku, Chiba 263-8555, Japan; 3Graduate School of Pharmaceutical Sciences, Hokkaido University, Kita-ku, Sapporo 060-0812, Japan; 4Department of Chemistry, Graduate School of Science, Kyoto University, Kitashirakawa-oiwakecho, Sakyo-ku, Kyoto 606-8502, Japan; 5Cyclotron and Radioisotope Center (CYRIC), Tohoku University, Sendai 980-8578, Japan; 6Research Center for Electron Photon Science (ELPH), Tohoku University, Sendai 982-0826, Japan; 7Institute for Integrated Cell-Material Science (iCeMS), Kyoto University, Yoshida-ushinomiyacho, Sakyo-ku, Kyoto 606-8501, Japan

**Keywords:** platinum-191, Auger electron, MYCN, neuroblastoma, cGAS-STING, interferon

## Abstract

Auger electrons can cause nanoscale physiochemical damage to specific DNA sites that play a key role in cancer cell survival. Radio-Pt is a promising Auger-electron source for damaging DNA efficiently because of its ability to bind to DNA. Considering that the cancer genome is maintained under abnormal gene amplification and expression, here, we developed a novel ^191^Pt-labeled agent based on pyrrole–imidazole polyamide (PIP), targeting the oncogene MYCN amplified in human neuroblastoma, and investigated its targeting ability and damaging effects. A conjugate of MYCN-targeting PIP and Cys-(Arg)_3_-coumarin was labeled with ^191^Pt via Cys (^191^Pt-MYCN-PIP) with a radiochemical purity of >99%. The binding potential of ^191^Pt-MYCN-PIP was evaluated via the gel electrophoretic mobility shift assay, suggesting that the radioagent bound to the DNA including the target sequence of the MYCN gene. In vitro assays using human neuroblastoma cells showed that ^191^Pt-MYCN-PIP bound to DNA efficiently and caused DNA damage, decreasing MYCN gene expression and MYCN signals in in situ hybridization analysis, as well as cell viability, especially in MYCN-amplified Kelly cells. ^191^Pt-MYCN-PIP also induced a substantial increase in cytosolic dsDNA granules and generated proinflammatory cytokines, IFN-α/β, in Kelly cells. Tumor uptake of intravenously injected ^191^Pt-MYCN-PIP was low and its delivery to tumors should be improved for therapeutic application. The present results provided a potential strategy, targeting the key oncogenes for cancer survival for Auger electron therapy.

## 1. Introduction

Radiation damages DNA, leading to cell death, gene mutation, and immune activation [1,2,3]. Auger electrons have low energy at the electron volt-to-kiloelectron volt scale, inducing nanoscale physiochemical damage to DNA [4]. When an Auger electron emitter is close to DNA, it damages the DNA effectively at the nanoscale [5,6]. A nanoscale radiation therapy targeting the key oncogenes involved in the survival mechanism of cancer can be realized with Auger-electron-emitting radioelements coupled with agents that recognize specific DNA sequences [7]; however, the development and application of such drugs remain elusive.

As one of the potential Auger-electron-emitting radionuclides for oncogene-targeted Auger electron therapy, we employed ^191^Pt (*T*_1/2_ = 2.80 d, EC = 100%, 13 Auger electrons/decay, 1 internal conversion electron/decay [8]) because of its high yields of Auger electrons/internal conversion electrons [8,9] and the superior DNA-binding ability of Pt [10]. This ability has realized the Pt-based antineoplastic drugs widely used to treat various types of cancer [10]. A previous study observed that ^191^Pt-labeled DNA intercalator Hoechst33258 exhibited better DNA-binding and damaging characteristics than ^111^In-labeled Hoechst33258 as a common radionuclide [11]. Taken together, ^191^Pt-labeled agents have a substantial advantage for use in DNA-targeting drugs. To advance therapeutic agents targeting key oncogenes, the delivery of ^191^Pt to a specific site of the genome in living cells is critical.

As a target in the genome, we focus on the oncogene MYCN. The MYCN gene is a transcription factor that is amplified in human neuroblastoma and is related to the patient’s prognosis [12,13]. The survival of cancer cells depends on such specific gene amplification and overexpression [14,15,16,17]. In addition, damaging these factors and regulators not only induces cell death but also activates the antitumor immune system via the cyclic GMP–AMP synthase–stimulator of interferon genes (cGAS–STING) pathway [18,19,20]. If Auger electrons damage such factors/regulators, they would cause additional biological immune responses after the damage. The MYCN gene is a promising target in Auger electron therapy for neuroblastoma.

To deliver ^191^Pt to a target gene, an appropriate module needs to be incorporated into a ^191^Pt-labeled compound. Pyrrole–imidazole polyamides (PIPs) are candidates for delivering ^191^Pt to a specific site of the genome. Numerous PIPs that target cancer-related genes have been developed, and some have been demonstrated to express therapeutic efficacy in preclinical studies with mice and marmosets [21,22]. In the case of the MYCN gene, a MYCN-targeting PIP developed by Yoda et al. showed a specific targeting ability for and therapeutic effects in MYCN-amplified cells [23]. This well-established MYCN-PIP can be a lead compound to demonstrate the concept of gene-targeted Auger electron therapy with ^191^Pt. 

We here designed a ^191^Pt-labeled conjugate, ^191^Pt-MYCN-PIP, based on the MYCN-targeting PIP as described above. The conjugate consists of the MYCN-targeting PIP [23] conjugated with cysteine (Cys) as the labeling site of ^191^Pt [11], tri-arginine (R3) as a cell penetration module [24], and the fluorescent compound coumarin (Figure 1). ^191^Pt-MYCN-PIP was synthesized, and its in vitro properties were evaluated in MYCN-amplified and -unamplified neuroblastoma models.

## 2. Results

### 2.1. Synthesis of ^191^Pt-Labeled PIP and the Evaluation of Its Target Sequence-Binding Ability, Cell Uptake, and DNA Binding

The MYCN-Cys-R3-coumarin (**1**, MYCN-PIP) and GCC-Cys-R3-coumarin (**2**, GCC-PIP) were synthesized as illustrated in Figure 1 and Appendix A. GCC-PIP has a half chain of the MYCN-targeting moiety as a low-affinity control. ^191^Pt-labeled MYCN/GCC-PIP was obtained in a high radiochemical yield of 50–70%. After purification via preparative high-performance liquid chromatography (HPLC), ^191^Pt-MYCN/GCC-PIP was obtained with a radiochemical purity greater than 95% at the end of synthesis, as shown in the analytical HPLC chromatogram (Figure 2a and Appendix A).

First, the MYCN-targeting ability of MYCN-PIP was determined, via gel electrophoretic mobility shift assay (EMSA), to be maintained after it was labeled with ^191^Pt. Figure 2b shows the fluorescence images of EtBr-stained oligonucleotides (lanes 1–5) and the autoradiograph of ^191^Pt radioactivity (lane 6). Untreated oligonucleotides were observed in a single band (lane 1), and treatment with GCC-PIP and MYCN-PIP generated mobility-shifted bands (lanes 2 and 3). MYCN-PIP (lane 3) resulted in a much more intense shifted band than GCC-PIP as the low-affinity control (lane 2), suggesting that MYCN-PIP exhibits a greater targeting ability than GCC-PIP. ^191^Pt-MYCN-PIP also generated the mobility-shifted bands observed for unlabeled MYCN-PIP (lanes 3, 5, and 6). In the autoradiograph, ^191^Pt radioactivity was observed in the shifted band but not in the unshifted original band (lane 6).

The cellular uptake and DNA-binding behavior of ^191^Pt-MYCN-PIP in cultured human neuroblastoma cells with different levels of MYCN gene amplification (amplified: Kelly; unamplified: SK-N-AS) were subsequently evaluated. The cellular uptake of ^191^Pt-MYCN-PIP was saturated within 3 h of incubation and was similar between the two cell lines (Figure 2c left). ^191^Pt-MYCN-PIP exhibited the greatest cellular uptake, followed by ^191^Pt-GCC-PIP and free ^191^Pt (Figure 2c right). The amount of ^191^Pt bound to DNA was approximately the same in both the Kelly and SK-N-AS cell lines with different copy numbers of the MYCN gene (Figure 2d). Both the cellular uptake and the DNA-binding fraction of ^191^Pt-MYCN-PIP were one order of magnitude greater than those of ^191^Pt-GCC-PIP. This result suggests that ^191^Pt-MYCN-PIP more readily moved into cells, was retained longer in the cells, and was more stably bound to DNA than ^191^Pt-GCC-PIP. To evaluate DNA double-strand breaks (DSBs) induced by ^191^Pt-MYCN-PIP, the DSB marker 53BP1 was fluorescently imaged (Appendix A). A substantial increase of the 53BP1-EGFP foci was observed in the ^191^Pt-MYCN-PIP treatment, compared with the treatment using MYCN-PIP; the damage was likely caused by Auger electrons emitted from ^191^Pt-MYCN-PIP.

### 2.2. Cytotoxicity and DNA-Damaging Effects of [^191^Pt]Pt-MYCN-Cys-R3-Coumarin

The cytotoxicity was evaluated via a cell viability assay using three different cell lines (amplified: Kelly, SK-N-DZ; unamplified: SK-N-AS) treated with ^191^Pt-MYCN-PIP, MYCN-PIP, or phosphate-buffered saline (PBS). The copy number of the MYCN gene in Kelly cells was greater than that in SK-N-DZ cells [25]. The cytotoxicity of ^191^Pt-MYCN-PIP differed among the three cell lines (Figure 3a). For highly MYCN-amplified Kelly cells, ^191^Pt-MYCN-PIP decreased cell viability substantially more than only nonradioactive MYCN-PIP or PBS: ^191^Pt-MYCN-PIP was 0.35 and 0.44, and MYCN-PIP was 0.66 and 0.90 (*n* = 2), when PBS was normalized as 1, although no significant difference was observed. A moderate effect of ^191^Pt-MYCN-PIP on slightly amplified SK-N-DZ cells and almost no effect of ^191^Pt-MYCN-PIP on unamplified SK-N-AS cells were observed compared with the effects of only nonradioactive MYCN-PIP or PBS (Figure 3a).

The quantitative reverse transcription PCR (RT-qPCR) analysis of the MYCN gene shows that the MYCN gene expression differed substantially among the cell lines (Figure 3b left). The highest expression was observed in Kelly cells, approximately one third of that expression was observed in SK-N-DZ cells, and little expression was observed in SK-N-AS cells. In all the cells, treatment with ^191^Pt-MYCN-PIP decreased the expression of MYCN compared with treatment with MYCN-PIP and PBS (Figure 3b right), suggesting that Auger-electron-emitting ^191^Pt affected MYCN gene expression. On the other hand, the MYCN gene expression tended to increase during the treatment with MYCN-PIP compared with the control (PBS treatment), although the mechanism for this was unclear. Our ^191^Pt-MYCN-PIP included non-radioactive MYCN-PIP, and the effect of ^191^Pt-MYCN-PIP might be masked by the effect of MYCN-PIP.

For another evaluation of the damaging effect of ^191^Pt-MYCN-PIP on the MYCN gene, fluorescence in situ hybridization (FISH) imaging was conducted in Kelly cells. In highly MYCN-amplified Kelly cells, many signals of the MYCN gene were observed (Figure 3c). Approximately the same number of signals observed in nuclei treated with PBS were also observed in nuclei treated with MYCN-PIP. In contrast, a significant decrease in MYCN signals was observed in nuclei treated with ^191^Pt-MYCN-PIP. These results are consistent with the previously discussed RT-qPCR results.

To evaluate DNA replication stress following damage to the transcription factor MYCN gene, we carried out immunostaining with anti-dsDNA. The cytosolic dsDNA released by cell damage and DNA cleavage can be recognized by anti-DNA and compose immune complexes [26]. Free ^191^Pt did not induce a change; the cells were similar to those in PBS (Figure 4a). MYCN-PIP and ^191^Pt-GCC-PIP resulted in a few cytosolic dsDNA granules outside of the cell nuclei. However, ^191^Pt-MYCN-PIP resulted in numerous cytosolic dsDNA granules, and the whole cell fluorescence signal was significantly stronger after the treatment with ^191^Pt-MYCN-PIP than after treatments with MYCN-PIP and ^191^Pt-GCC-PIP. Such cytosolic dsDNA granules are known to induce type I interferons such as IFN-α and IFN-β in the activated cGAS-STING pathway [27,28]. Our RT-qPCR analysis also found more IFN-α and IFN-β released in Kelly cells treated with ^191^Pt-MYCN-PIP than in Kelly cells treated with PBS or nonradioactive MYCN-PIP (Figure 4b).

### 2.3. Biodistribution of [^191^Pt]Pt-MYCN-Cys-R3-Coumarin in Mice Bearing Neuroblastoma Tumors

The biodistribution of ^191^Pt-MYCN-PIP was evaluated in a xenograft mouse model bearing Kelly tumors after intravenous injection, represented in Appendix A (lung, liver, and spleen) and Appendix A (blood, brain, intestine, kidney, muscle, bone, and tumor). A high accumulation of ^191^Pt-MYCN-PIP in the lungs was observed in the early time points; the peak was 342%IA/g at 2 min after injection (Appendix A). The accumulation in the spleen increased with time until 1 d (100%IA/g), and relatively high hepatic accumulation was observed; the peak was 26%IA/g at 1 d (Appendix A). The red blood cell (RBC) partitioning proceeded rapidly after the injection; the rate was 85% at 2 min and 55% thereafter at 1–4 d (Appendix A). In contrast, the plasma protein binding rate was 50% at 2 min and increased to 83% at 1 d after injection (Appendix A). Consequently, little accumulation of ^191^Pt-MYCN-PIP was observed in the tumors: less than 1%IA/g (Appendix A). On the other hand, in the in vivo evaluation with intratumoral injection, ^191^Pt-MYCN-PIP was up-taken into tumor cells and bound to DNA (Appendix A). Most of the intratumorally injected ^191^Pt-MYCN-PIP was retained in the Kelly tumors and not excreted (357%IA/g at 4 d) (Appendix A). The in vivo DNA-binding rate and the DNA-binding-to-cell-uptake ratio were comparable to those in the in vitro experiments (Appendix A).

## 3. Discussion

The present study demonstrated that a ^191^Pt-labeled agent based on MYCN-targeting PIP expressed cytotoxicity in MYCN-amplified cells, the survival of which depended on the MYCN gene expression, suggesting oncogene targeting would be a promising strategy for Auger electron therapy. The Auger-electron-emitting ^191^Pt-MYCN-PIP was successfully synthesized with high radiochemical yields and purity (Figure 2a). ^191^Pt-MYCN-PIP targeted the MYCN gene in living cells, decreasing gene expression and gene signals (Figure 3b,c). Substantial cytotoxicity was observed in MYCN-amplified neuroblastoma Kelly cells but not in unamplified SK-N-AS cells (Figure 3a) because MYCN plays a key role in the survival of MYCN-amplified neuroblastoma cells but not in unamplified ones [12,13] Additionally, a significant increase in DNA damage and replication stress was observed, as indicated by cytosolic dsDNA granules and IFN-α and IFN-β (Figure 4a,b). Unfortunately, the tumor uptake of intravenously injected ^191^Pt-MYCN-PIP was low. Although its delivery to tumors should be improved for in vivo therapeutic application, the present research has found a potential strategy for targeting key oncogenes in cancer survival for Auger electron therapy.

High radiochemical yields and purities were achieved for the current ^191^Pt-labeled compounds (Figure 2a). In a previous study, we obtained only a low radiochemical yield and purity for another ^191^Pt-labeled compound via Cys because of the high reactivity of Pt toward the thiol group and the multivalent coordination of ^191^Pt [11]. However, in the present study, changing the position of Cys in the compounds led to high radiochemical yields and purities for ^191^Pt-MYCN-PIP and ^191^Pt-GCC-PIP. The Cys was positioned next to the PIP molecule and R3; these middle molecules might create steric hindrance and suppress the multivalent coordination of ^191^Pt. The current design is suitable for further drug development with radio-Pt.

The targeting ability of MYCN-PIP was maintained after ^191^Pt-radiolabeling. EMSA with oligonucleotides containing the target sequence showed that ^191^Pt-MYCN-PIP generated mobility-shifted bands like unlabeled MYCN-PIP’s (Figure 2b, lanes 3, 5, and 6). There was no radioactivity of ^191^Pt at the unshifted band of the oligonucleotides, suggesting that ^191^Pt-MYCN-PIP maintained the MYCN-targeting ability after labeling (Figure 2b). The mobility-shifted band was observed for ^191^Pt-MYCN-PIP more than ^191^Pt-GCC-PIP. The DNA extraction assay also showed that the DNA-binding ability of ^191^Pt-MYCN-PIP was greater than that of ^191^Pt-GCC-PIP, as expected from the compound design (Figure 2d). Unfortunately, considering there were no differences in the cell uptake and DNA binding of ^191^Pt-MYCN-PIP between MYCN-amplified Kelly and nonamplified SK-N-AS cell lines (Figure 2c,d), ^191^Pt-MYCN-PIP could bind to other genomic regions in addition to the target sequence.

Irrespective of certain concerns about off-target effects, there was a relationship between the cytotoxicity of ^191^Pt-MYCN-PIP and the copy number of the MYCN gene in neuroblastoma cell lines. The cytotoxicity of ^191^Pt-MYCN-PIP was greater in Kelly cells (highly amplified) than in SK-N-DZ (moderately amplified) and SK-N-AS (unamplified) cells (Figure 3a). In MYCN-amplified Kelly cells, ^191^Pt-MYCN-PIP decreased the MYCN signals in FISH imaging and the MYCN gene expression in RT-qPCR (Figure 3b,c). These results suggest that ^191^Pt-MYCN-PIP damaged the target MYCN gene, and that cytotoxicity could depend on the role of MYCN in each cell line. Kelly cells have one of the highest expressions of MYCN in human neuroblastoma cell lines [25], and their survival hallmark depends on MYCN-related transcription [12,13]. Kang et al. reported that RNA interference targeting MYCN inhibited cell proliferation and caused cell death in the MYCN-amplified cells more than non-amplified cells [29]. Considering our results alongside other researchers’ findings, the different cytotoxicities are most likely related to the MYCN dependency: the copy number and expression of the MYCN gene.

The immunofluorescent staining of dsDNA and the PCR assay for IFN-α/β suggest that ^191^Pt-MYCN-PIP induced DNA fragmentation and genomic DNA conformation change, leading to a secondary damage response. Cytosolic dsDNA granules induce IFN production via cGAS-STING activation [27,28]. ^191^Pt-MYCN-PIP increased cytosolic dsDNA granules, resulting in a subsequent increase in IFN-α and IFN-β in Kelly cells. This result means that ^191^Pt-MYCN-PIP induced DNA replication stress and activated a cytosolic dsDNA-related pathway like the cGAS-STING pathway by damaging the MYCN gene and its downstream genes, which maintain DNA replication and genome stability in MYCN-amplified cells [22,23]. In addition to cytosolic dsDNA, the intensity of nucleic dsDNA also increased. This tendency could be because of increased antibody accessibility to nucleic dsDNA due to DNA fragmentation and genomic structure alteration. Such DNA conformation change might also be involved in an increase in IFN-α/β expression. Hence, ^191^Pt-MYCN-PIP likely damaged the genome integrity and activated a response pathway such as cGAS-STING.

The biodistribution of ^191^Pt-MYCN-PIP suggests that the compound design should be improved to enable its delivery to tumors via intravenous injection. According to previous research on the biodistribution of ^18^F-labeled PIP compounds, PIP compounds are primarily biodistributed to the liver [30]. However, our ^191^Pt-MYCN-PIP was mainly accumulated in the lungs at the early time points and the spleen at the late time points (Appendix A). This biodistribution pattern of ^191^Pt-MYCN-PIP was also different from that of free ^191^Pt. The high accumulation of ^191^Pt-MYCN-PIP in the lungs is likely due to acute pulmonary embolism (PE) [31,32,33,34]. High hydrophilic and cationic poly-bases readily interact with plasma proteins or the membrane of RBCs, inducing their agglutination and an acute PE. Thus, it was suggested that R3-coumarin could affect the biodistribution of the original MYCN-targeting PIP adversely, considering that the RBC partitioning and plasma protein binding of ^191^Pt-MYCN-PIP proceeded rapidly after intravenous injection (Appendix A). The compound design around R3-coumarin needs to be redesigned. There is a recent report that small, interfering, MYCN-targeting RNA combined with an RGD ligand worked in mice bearing Kelly tumors [35]. This encourages further developments in PIP-based radioagents by replacing R3-coumarin with tumor-targeting modules.

## 4. Materials and Methods

### 4.1. General

Chemicals and reagents were purchased from standard suppliers: FUJIFILM Wako Pure Chemical (Osaka, Japan), Nacalai Tesque (Kyoto, Japan), Tokyo Chemical Industry (Tokyo, Japan), Watanabe Chemical Industries (Hiroshima, Japan), Peptide Institute (Osaka, Japan), Otsuka Pharmaceutical Factory (Tokyo, Japan), and Merck (Darmstadt, Germany). Milli-Q ultrapure water or distilled water was used for dilution in all experiments.

High-purity germanium (HPGe) γ-ray spectrometry was performed to measure the radioactivity of ^191^Pt before all in vitro/in vivo experiments. The HPGe detector (EGC 15–185-R; Eurisys Measures, Strasbourg, France) was coupled with a 4096 multichannel analyzer (RZMCA; Laboratory Equipment, Inashiki, Japan). A gamma counter (Wizard2 2-Detector Gamma Counter; PerkinElmer, Waltham, MA, USA) was used to measure radioactivity in biological samples.

### 4.2. Synthesis of MYCN-Cys-R3-Coumarin (***1***) and GCC-Cys-R3-Coumarin (***2***)

Each fluorenylmethyloxycarbonyl (Fmoc) monomer unit (Fmoc-d-Arg(Pbf)-OH, Fmoc-l-Cys(Trt)-OH, Fmoc-β-alanine-OH, Fmoc-γ-butyl-OH, Fmoc-*N*-methylpyrrole(Py)-OH, and Fmoc-*N*-methylimidazole(Im)-OH) was introduced to Fmoc-d-Arg(Pbf)-Alko resin (0.59 mmol/g, 100–200 mesh) using a PSSM-8 automated peptide synthesizer (Shimadzu, Kyoto, Japan). Each coupling reaction was performed at room temperature (rt) for 1 h in *N*-methyl-2-pyrrolidone (NMP) containing 4 equiv. of *O*-(1*H*-6-chlorobenzotriazole-1-yl)-1,1,3,3-tetramethyluronium hexafluorophosphate (HCTU) and *N*,*N*-diisopropylethylamine (DIEA), followed by Fmoc deprotection in a solution of 20% piperidine in *N*,*N*-dimethylformamide (DMF). After solid-phase synthesis, the resin was incubated with 20% acetic anhydride in DMF at rt for acetyl capping.

The Py–Im polyamide (PIP) compound was cleaved with 3,3′-diamino-*N*-methyldipropylamine at 55 °C for 3 h and precipitated in Et_2_O. The supernatant Et_2_O was removed after centrifugation, and the resultant crude powder was dried in vacuo. The crude product was coupled with 2 equiv. of 7-methoxycoumarin-3-carboxylic acid in DMF (0.34 mL) by adding 2 equiv. of benzotriazole-1-yl-oxy-tris-pyrrolidino-phosphonium hexafluorophosphate (PyBOP) and 4 equiv. of DIEA and shaking the resultant mixture at rt for 2 h. The reaction solution was added into Et_2_O, the crude product was precipitated, and the obtained solid was dried in vacuo. The Pbf and Trt groups of amino residues were removed via treatment with a TFA:triisopropylsilane:water (1 mL = 95:2.5:2.5 *v*/*v*%) solution at rt for 1 h. The reaction solution was drained into Et_2_O, and the resultant solid was dried in vacuo to purify on a CombiFlash Rf RFJ model equipped with a RediSep Rf 4.3 g C18 reverse-phase column (Teledyne ISCO, Lincoln, NE, USA). The eluent was A/0.1% trifluoroacetic acid (TFA) in H_2_O, B/0.1% TFA in CH_3_CN, gradient: A/B = 100/0 in 0 min → 65/35 in 35 min.

HPLC analysis was performed on a Jasco Engineering PU-2089 Plus system equipped with a COSMOSIL 150 × 4.6 mm^2^ 5C_18_-MS-II packed column (Nacalai Tesque, Kyoto, Japan); the eluent was A/0.1% TFA in water, B/acetonitrile, a linear gradient elution of 0–75% acetonitrile over 30 min was used; the flow rate was 1.0 mL/min; the detection was at 254 nm. Collected fractions were analyzed via matrix-assisted laser desorption/ionization time-of-flight mass spectrometry (MALDI-TOF-MS) using a Microflex-KS II (Bruker, Billerica, MA, USA).

MYCN-PIP (**1**): Analytical HPLC: *t*_R_ = 17.0 min. MALDI-TOF MS: *m*/*z* calcd. for C_120_H_157_N_50_O_25_S^+^ [M+H]^+^ 2730.2, found 2730.6. GCC-PIP (**2**): Analytical HPLC: *t*_R_ = 15.4 min. MALDI-TOF MS: *m*/*z* calcd. for C_81_H_114_N_33_O_17_S^+^ [M+H]^+^ 1852.9, found 1853.4 (Appendix A). MYCN-PIP and GCC-PIP were >95% pure, as determined via HPLC.

### 4.3. Synthesis of [^191^Pt]Pt-MYCN-Cys-R3-Coumarin and [^191^Pt]Pt-GCC-Cys-R3-Coumarin

Two hundred and thirty microliters of the reaction mixture consisting of dimethyl sulfoxide (DMSO) (76 µL), ^191^Pt solution (3.7–93 MBq, 150 µL), 26 µM MYCN/GCC-Cys-R3-coumarin (2 mM in DMSO, 3 µL), 0.09% Tween 80 (0.2 µL), and 0.3% ethanol (0.8 µL) was prepared and heated at 45 °C for 60 min. The reaction mixture was purified via HPLC (injected volume: 230 µL); the column was a COSMOSIL 5C18-MS-II (5 µm, 4.6 mm × 250 mm; Nacalai Tesque, Kyoto, Japan), the eluent was A/0.1% TFA in H_2_O, B/0.1% TFA in CH_3_CN, gradient: A/B = 95/5 in 0 min → 30/70 in 25 min, and the flow rate was 1.0 mL/min. The HPLC system (PU-4080i, MD-4010; Jasco, Tokyo, Japan) was equipped with radiation detectors (US-3000; Universal Giken, Odawara, Japan and model 2200/44-10; Ludlum Measurements, Sweetwater, TX, USA). The radiochemical yield was evaluated as the radioactivity ratio of the purified product to the crude sample injected onto the preparative HPLC. The radioactivity was measured using a curiemeter (IGC-7R, Aloka, Japan).

A solution containing [^191^Pt]Pt-MYCN/GCC-Cys-R3-coumarin was collected in a vial with Tween 80 added beforehand. The solution was evaporated to remove CH_3_CN and neutralized with 10× PBS and 1 M NaOH. [^191^Pt]Pt-MYCN-Cys-R3-coumarin was finally prepared in a 1× PBS solution containing 0.2–0.4% Tween 80 and 0.2% TFA for in vitro experiments or containing 0.75% Tween 80 and 0.2% TFA for in vivo experiments. After purification, the product was analyzed via HPLC (injected volume: 100 µL) under the same conditions used in preparative HPLC, and the radiochemical purity was determined. Because the product solution of ^191^Pt-MYCN-PIP contains nonlabelled MYCN-PIP, the following in vitro/in vivo experiments involving the treatment of nonlabelled MYCN-PIP with approximately the same concentration as the ^191^Pt-MYCN-PIP solution (1–20 µmol/L for an incubation or injection solution) were conducted for comparison.

### 4.4. Gel Electrophoretic Mobility Shift Assay (EMSA)

Oligonucleotides (30 bp: forward, 5′-CCTAACCACAAGA***AGGCTCCCA***AGTTAACC-3′; reverse, 3′-GGATTGGTGTTCT***TCCGAGGGT***TCAATTGG-5′) were designed and obtained from Sigma-Aldrich. The oligonucleotides were dissolved in Tris buffer (10 mM, pH = 7.5) at a concentration of 100 µM and were formed as double strands by heating at 90 °C for 5 min. The oligonucleotide samples (32 µL) were prepared at a DNA concentration of 5 µM in a buffer solution containing 1% DMSO, 0.5% Tween 80, 10 mmol/L (CH_3_)_2_AsO_2_Na, 10 mmol/L Tris-HCl, 10 mmol/L NaCl, and nonradioactive GCC/MYCN-PIP (30–55 µmol/L) with/without ^191^Pt-MYCN-PIP (90 kBq). After the samples were mixed, they were incubated at 37 °C overnight (15 h). Samples of only ^191^Pt-MYCN-PIP without oligonucleotides or only oligonucleotides were also prepared as a control in the same buffer. Ten microliters of sample solution were mixed with 2 μL of loading dye (Gel Loading Dye, Purple 6×, New England Biolabs, Ipswich, MA, USA) and loaded into a gel lane. The gel purchased from ATTO (15%/e-PAGEL; Tokyo, Japan) was run in 1× TG buffer (0.025 mol/L Tris base and 0.192 mol/L glycines) at 20 mA for 60 min at 4 °C and then stained with 2.5 µg/mL ethidium bromide (EtBr) in 0.5× TBE buffer solution (0.0445 mol/L Tris base, 0.0445 mol/L borates, and 0.001 mol/L EDTA). The EtBr-stained gel was scanned on an image analyzer (ImageQuant LAS 500; Cytiva, Tokyo, Japan). After the gel was dried, its autoradiograph was obtained using an imaging plate.

### 4.5. In Vitro Assays of Cultured Cells

The human MYCN-amplified neuroblastoma cell lines Kelly and SK-N-DZ and the unamplified cell line (one copy) SK-N-AS were used. A cellular uptake assay, DNA-binding assay, cytotoxicity assay, RT-qPCR analysis of MYCN and cytokines, FISH for analysis of MYCN, and immunostaining for analysis of dsDNA and γH2AX were conducted.

#### 4.5.1. Cellular Uptake

Kelly or SK-N-AS cells ((8–9) × 10^5^ cells/6-well, 1 × 10^5^ cells/24-well) were seeded onto a 6-well (1 mL media) or 24-well (0.5 mL media) plate and incubated at 37 °C in a humidified atmosphere containing 5% CO_2_. No carrier-added free ^191^Pt, ^191^Pt-GCC-PIP, or ^191^Pt-MYCN-PIP (80–90 kBq), was added to the cells (nonradioactive MYCN-PIP or GCC-PIP: 0.03 nmol/24-well, 1 nmol/6-well). After the cells had incubated for 3–42 h, they were washed with PBS and dissolved in 0.1 M NaOH. The radioactivity was measured using a gamma counter, and the protein concentration was determined using the Bradford reagent (Quick Start Bradford Protein Assay Kit; Bio-Rad Laboratories, Hercules, CA, USA) for calculation of the uptake value.

#### 4.5.2. DNA-Binding Assay

Kelly or SK-N-AS cells ((8–10) × 10^5^ cells/well) were seeded onto a 6-well plate and incubated (1 mL media). ^191^Pt-MYCN-PIP (80 kBq) or ^191^Pt-GCC-PIP (65 kBq) was added to the cells (nonradioactive MYCN-PIP or GCC-PIP: 0.1 nmol/well). After the cells were incubated for 1 d (22 h), genomic DNA was isolated from the cells using a NucleoSpin Tissue DNA Extraction Kit (MACHEREY-NAGEL, Düren, Germany). The radioactivity of the collected solution containing genomic DNA was measured using a gamma counter (Wizard2 2-Detector Gamma Counter; PerkinElmer, Waltham, MA, USA), and quantitative analysis was performed using a NanoDrop One microvolume UV–Vis spectrophotometer (Thermo Fisher Scientific, Waltham, MA, USA). The DNA-binding rate was calculated on the basis of the measured values.

#### 4.5.3. Cell Viability Assay

Kelly, SK-N-AS, and SK-N-DZ (1 × 10^5^ cells/well) were seeded onto a collagen I-coated 24-well plate (Corning, Corning, NY, USA) and incubated (0.5 mL media). ^191^Pt-MYCN-PIP (457 kBq) + nonradioactive MYCN-PIP (0.05 nmol), only nonradioactive MYCN-PIP (0.05 nmol), or PBS, was added to the cells, and the cells were incubated for 1.5 d (39–40 h). After the medium was removed and the cells were washed with PBS, the cells were stained in a reduced-serum medium (Opti-MEM; Thermo Fisher Scientific, Waltham, MA, USA) containing calcein (LIVE/DEAD Viability/Cytotoxicity Kit for mammalian cells; Thermo Fisher Scientific). The fluorescence was measured using a microplate reader (Spectra Max M5; Molecular Devices, Tokyo, Japan).

#### 4.5.4. Quantitative Reverse Transcription PCR (RT-qPCR)

Kelly, SK-N-DZ (1 × 10^5^ cells/24-well), and SK-N-AS (1 × 10^6^ cells/6-well) were seeded onto a 24-well (0.5 mL media) or 6-well (1 mL media) plate and incubated. ^191^Pt-MYCN-PIP (457 kBq for 24-well, 732 kBq for 6-well) + nonradioactive MYCN-PIP (0.05 nmol for 24-well, 0.08 nmol for 6-well), only nonradioactive MYCN-PIP (0.05 nmol for 24-well, 0.08 nmol for 6-well), or PBS, was added to the cells, and the cells were incubated for 2 d (42–50 h). After the medium was removed, the cells were dissolved in lysis buffer. Complementary DNA (cDNA) was prepared using a Fastlane Cell cDNA Kit (QIAGEN, Limburg, The Netherlands) from the cell lysates. The quantitative PCR for the synthesized cDNA was performed using a StepOne Real-Time PCR system (Applied Biosystems, Thermo Fisher Scientific). TaqMan Gene Expression assays (MYCN, Hs00232074_m1; IFNα1, Hs00256882_s1; IFNBβ1, Hs01077958_s1; Thermo Fisher Scientific) and Eukaryotic 18S rRNA Endogenous Control (VIC™ probe; Applied Biosystems, Thermo Fisher Scientific) were used for RT-qPCR, and quantitative analysis was based on the comparative cycle threshold (CT) method (ΔΔCT method).

#### 4.5.5. Fluorescence In Situ Hybridization (FISH) Analysis

Kelly cells (6 × 10^4^) were seeded onto a slide chamber (Corning^®^ BioCoat^®^ Collagen I 8-well Culture Slide; Corning, Corning, NY, USA) and incubated (0.3 mL media). ^191^Pt-MYCN-PIP (274 kBq) + nonradioactive MYCN-PIP (0.03 nmol), only nonradioactive MYCN-PIP (0.03 nmol), or PBS, was added to the cells, and the cells were incubated for 1.5 d (38 h). After the medium was removed and the cells were washed with PBS, the cells were immobilized with ice-cold Carnoy solution (acetic acid/methanol = 1/3). The immobilized cells were denatured with a denaturing solution containing 70% formamide and 2× saline sodium citrate buffer (SSC) and then dehydrated with 70, 85, and 100% ethanol. The prepared cells were treated with a MYCN FISH probe solution for hybridization (MYCN FISH Probe Red-dUTP; Empire Genomics, Buffalo, NY, USA) at 37 °C for 16 h in a humidified atmosphere containing 5% CO_2_. Cover glasses were placed on glass slides using a mounting medium containing DAPI (Vector Laboratories, Burlingame, CA, USA). Fluorescence images were taken with a BZ-X800 fluorescence microscope (KEYENCE, Osaka, Japan) and analyzed using the BZ-Z800 Analyzer software 1.1.2.4 (KEYENCE). Data were obtained from five different images with 30−80 nuclei per image and are expressed as the ratio of total fluorescence intensity of the MYCN gene to the nucleus area.

#### 4.5.6. Immunofluorescence Staining of Double-Stranded DNA (dsDNA)

Kelly cells (1 × 10^6^) were seeded on coverslips (BioCoat collagen I 22 mm coverslips; Corning, Corning, NY, USA) and incubated (1.5 mL media). ^191^Pt-MYCN-PIP (836 kBq) + nonradioactive MYCN-PIP (0.1 nmol), ^191^Pt-GCC-PIP (872 kBq) + nonradioactive GCC-PIP (0.1 nmol), free ^191^Pt (899 kBq), only nonradioactive MYCN-PIP (0.1 nmol), or PBS, was added to the cells, and the cells were incubated for 2 d (45 h). After the medium was removed and the cells were washed with PBS, the cells were immobilized with ice-cold MeOH, incubated in 0.25% Triton X-100/PBS for permeabilization, and then incubated in 1% BSA/PBST (phosphate-buffered saline with Tween 20) for blocking. Immunofluorescence staining was conducted using a dsDNA antibody (ab27156; Abcam, Cambridge, UK) as the primary antibody, with Alexa Fluor 594 anti-mouse IgG (Thermo Fisher Scientific) as the secondary antibody. Coverslips were placed on glass slides using a mounting medium containing DAPI (Vector Laboratories, Burlingame, CA, USA). Fluorescence images were acquired with a BZ-X800 fluorescence microscope (KEYENCE, Osaka, Japan) and analyzed using the BZ-Z800 Analyzer software 1.1.2.4 (KEYENCE). Data were obtained from four different images with 20−60 nuclei per image and are expressed as the ratio of total fluorescence intensity of dsDNA to the nucleus area.

### 4.6. Statistical Analysis

The data are expressed as means ± standard deviation (SD). The data were evaluated via one-way analysis of variance with multiple comparisons using the GraphPad Prism 9 software (ver. 9.0.2, GraphPad Software, San Diego, CA, USA); *p* < 0.05 was considered statistically significant. Each dot indicates a value per image. ns: *p* ≥ 0.05, *: *p* < 0.05, **: *p* < 0.01, ***: *p* < 0.001, ****: *p* < 0.0001.

## 5. Conclusions

The ^191^Pt-labeled compound ^191^Pt-MYCN-PIP targeting the oncogene MYCN was developed. Irrespective of no differences in the cell uptake and DNA binding of ^191^Pt-MYCN-PIP between MYCN-amplified Kelly and nonamplified SK-N-AS cells, ^191^Pt-MYCN-PIP decreased MYCN gene expression and MYCN signals in the FISH analysis in MYCN-amplified Kelly cells. Additionally, the agent induced cell toxicity in MYCN-amplified neuroblastoma cells but not in unamplified cells. An increase in DNA replication stress, as indicated by dsDNA granules, and subsequent IFN signaling activation were observed. Considering that the cancer genome is maintained under abnormal gene amplification and expression, Auger-electron-emitting drugs that target key oncogenes for cancer cell survival would be valid as a novel therapy for various tumor types. The tumor uptake of intravenously injected ^191^Pt-MYCN-PIP was very low, and its delivery to tumors should be improved for further therapeutic application.

## Figures and Tables

**Figure 1 pharmaceuticals-16-01526-f001:**
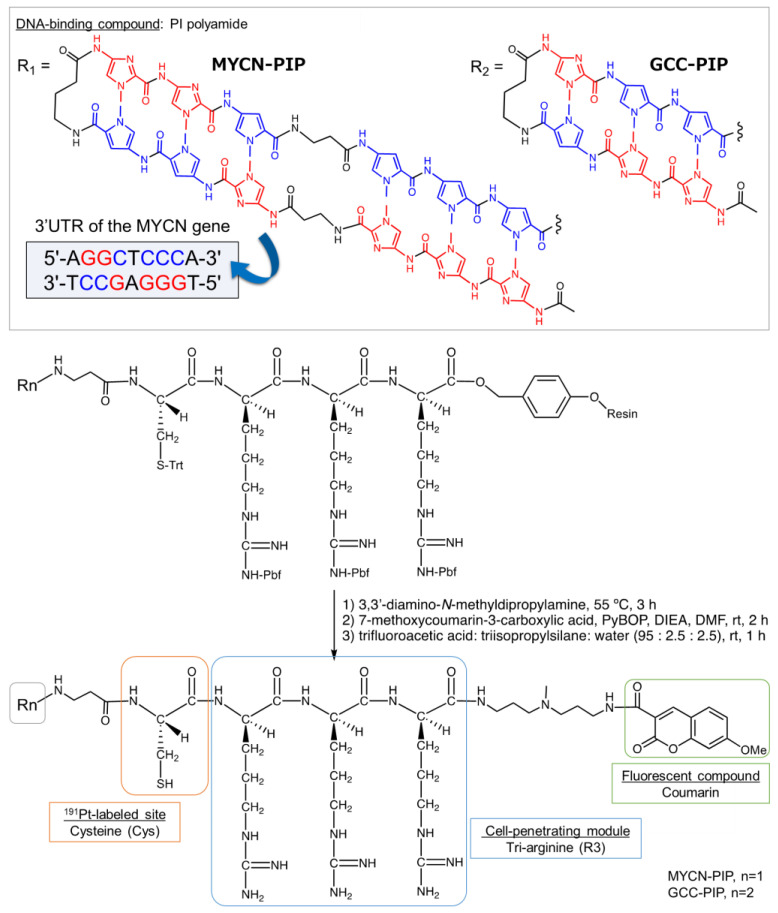
Synthesis of MYCN-PIP (**1**) and GCC-PIP (**2**).

**Figure 2 pharmaceuticals-16-01526-f002:**
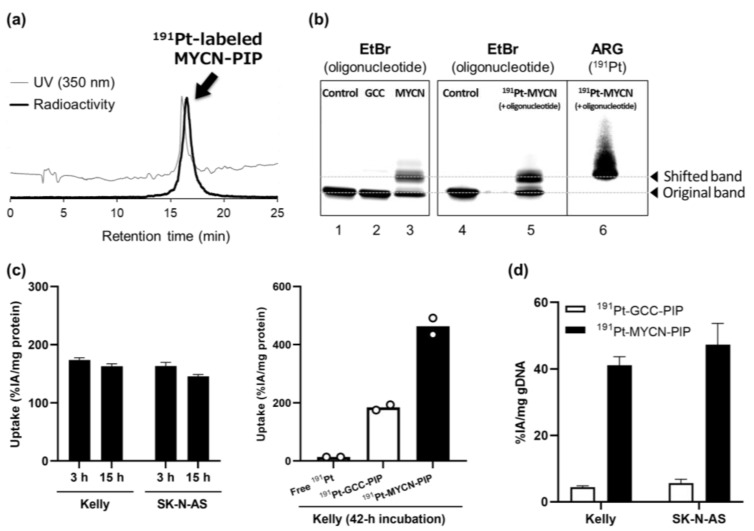
(**a**) HPLC radio chromatogram for the production of ^191^Pt-MYCN-PIP. (**b**) EMSA of oligonucleotides treated with no agent (control), nonradioactive GCC-PIP (GCC), nonradioactive MYCN-PIP (MYCN), or ^191^Pt-MYCN-PIP and nonradioactive MYCN-PIP (^191^Pt-MYCN). Lanes 5 and 6 of the same experiment and sample were detected via fluorescence and radioactivity, respectively. (**c**) Cellular uptake (% of the incubated activity (%IA)/mg protein) of ^191^Pt-MYCN-PIP (80–90 kBq/mL + nonradioactive MYCN-PIP (1 nmol/well) in Kelly and SK-N-AS cells after 3 and 15 h of incubation ((8–9) × 10^5^ cells/6-wells were seeded 1 day before the assay; *n* = 3, left). Cellular uptake (%IA/mg protein) of free ^191^Pt (160–180 kBq/mL), ^191^Pt-GCC-PIP + nonradioactive GCC-PIP (0.03 nmol/well), and ^191^Pt-MYCN-PIP (160–180 kBq/mL) + nonradioactive MYCN-PIP (0.03 nmol/well) in Kelly cells after 42 h of incubation (1 × 10^5^ cells/24-wells were seeded 1 day before the assay; *n* = 2, right). (**d**) DNA-binding rate in Kelly cells after 1 d of incubation (%IA/mg genomic DNA): ^191^Pt-GCC-PIP (80 kBq/mL) + nonradioactive GCC-PIP (0.1 nmol/well) and ^191^Pt-MYCN-PIP (65 kBq/mL) + nonradioactive MYCN-PIP (0.1 nmol/well) (*n* = 3).

**Figure 3 pharmaceuticals-16-01526-f003:**
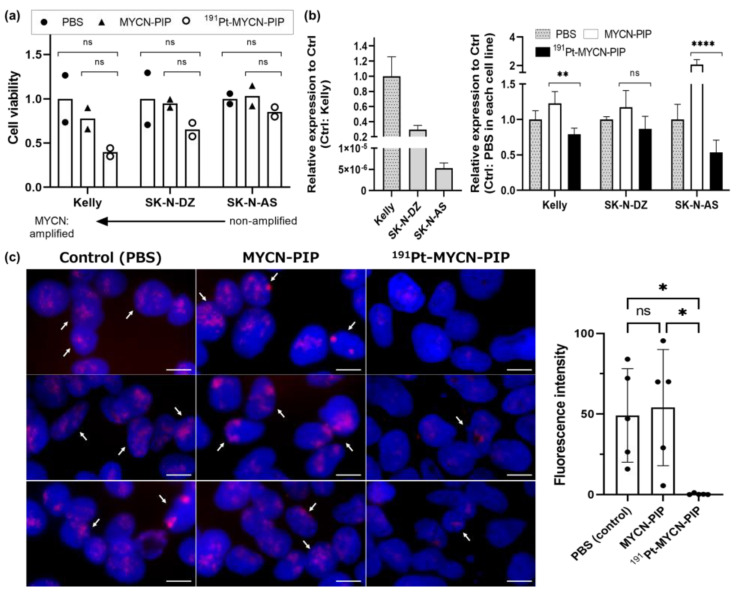
(**a**) Cell viability evaluated via live-cell staining. ^191^Pt-MYCN-PIP (457 kBq) + nonradioactive MYCN-PIP (0.05 nmol) (white circle), only nonradioactive MYCN-PIP (0.05 nmol) (black triangle), or PBS (black circle), was added to the cells, and the cells were incubated for 1.5 d (*n* = 2). No significant difference was observed. (**b**) Relative expression of the MYCN gene in RT-qPCR for MYCN in Kelly, SK-N-DZ, and SK-N-AS cells. Left: PBS was added to the cells, right: ^191^Pt-MYCN-PIP (457 kBq for 24-well, 732 kBq for 6-well) + nonradioactive MYCN-PIP (0.05 nmol for 24-well, 0.08 nmol for 6-well), only nonradioactive MYCN-PIP (0.05 nmol for 24-well, 0.08 nmol for 6-well), or PBS, was added to the cells, and the cells were incubated for 2 d. Controls were defined as relative expression = 1 (left, Ctrl = Kelly; right, Ctrl = PBS in each cell line) and the control columns were filled with dots. Data were averages of four replications, not significant (ns): *p* ≥ 0.05, **: *p* < 0.01, ****: *p* < 0.0001. (**c**) Representative fluorescence images of FISH imaging in Kelly cells. ^191^Pt-MYCN-PIP (274 kBq) + nonradioactive MYCN-PIP (0.03 nmol), only nonradioactive MYCN-PIP (0.03 nmol), or PBS, was added to the cells, and the cells were incubated for 1.5 d. Cell nuclei are indicated in blue (DAPI), and the MYCN gene is indicated in red. Scale bar: 10 µm. The quantitative data are expressed as the ratio of total fluorescence intensity of the MYCN gene to the nucleus area. ns: *p* ≥ 0.05, *: *p* < 0.05.

**Figure 4 pharmaceuticals-16-01526-f004:**
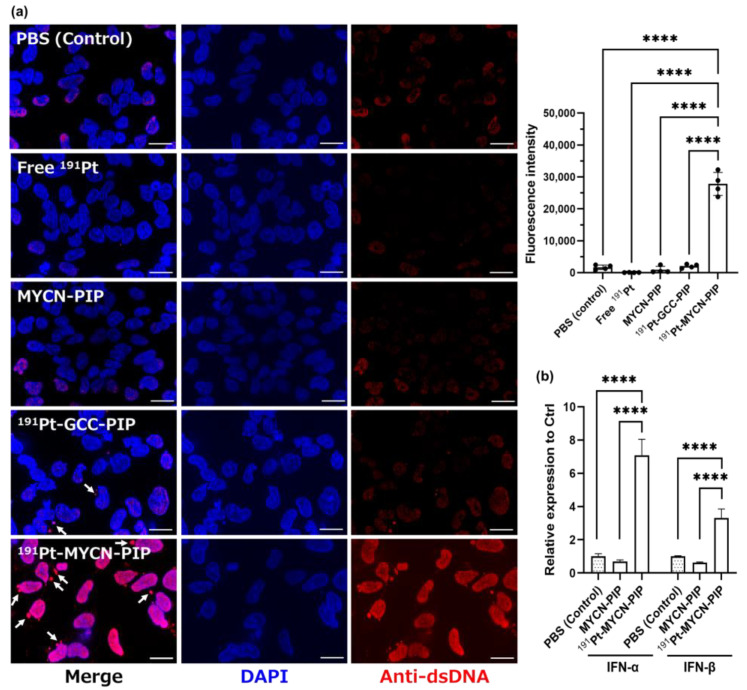
(**a**) Representative images of immunofluorescent staining of double-strand DNA (dsDNA) in Kelly cells. ^191^Pt-MYCN-PIP (836 kBq) + nonradioactive MYCN-PIP (0.1 nmol), ^191^Pt-GCC-PIP (872 kBq) + nonradioactive GCC-PIP (0.1 nmol), free ^191^Pt (899 kBq), only nonradioactive MYCN-PIP (0.1 nmol), or PBS, was added to the cells, and the cells were incubated for 2 d. Cell nuclei are indicated in blue (DAPI), and dsDNA granules are indicated in red. Scale bar: 20 µm. Arrows show the cytosolic dsDNA. The quantitative data are expressed as the ratio of total fluorescence intensity of dsDNA to the nucleus area. **** *p* < 0.0001. (**b**) Relative expression of the MYCN gene in RT-qPCR for IFN-α and IFN-β in Kelly cells. The PBS treatment is defined as the control (relative expression = 1) and the control columns were filled with dots. ^191^Pt-MYCN-PIP (457 kBq for 24-well) + nonradioactive MYCN-PIP (0.05 nmol for 24-well), only nonradioactive MYCN-PIP (0.05 nmol for 24-well), or PBS, was added to the cells, and the cells were incubated for 2 d (*n* = 4). **** *p* < 0.0001.

## Data Availability

Data is contained within the article and Appendix A.

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
