# Peer review of "Novel Auger-Electron-Emitting 191Pt-Labeled Pyrrole–Imidazole Polyamide Targeting MYCN Increases Cytotoxicity and Cytosolic dsDNA Granules in MYCN-Amplified Neuroblastoma"

_pharmaceuticals, 2023, doi:10.3390/ph16111526_

Round 1

Reviewer 1 Report

Comments and Suggestions for Authors

Recent developments in biologically-targeted radiotherapies showed a switch from traditional beta-emitters (90Y, 177LU) to the radionuclides with higher LET such as alpha-emitters and Auger electrons emitters (AEs). The submitted manuscript represents a very comprehensive study aimed to develop a new approach focused on gene-targeted Auger electron therapy using 191Pt as a radionuclide. The choice of pyrrole–imidazole polyamides (PIPs) as a targeting vehicle for cancer-related genes of neuroblastoma is well motivated as well as the choice of 191Pt as Auger emitter. The authors applied all the modern arsenal of in-vitro methods to evaluate new radiotherapeutic agent, [191Pt]Pt-MYCN-Cys-R3-coumarin, for cellular uptake and DNA-binding behavior, including cytotoxicity and DNA-damaging effects using three different cell lines. The results of all these in-vitro experiments were promising. However, in-vivo studies of the biodistribution of 191Pt-MYCN-PIP revealed that after intravenous (i.v.) injection of this new agent only less than 1 % of the injected activity (IA/g) was accumulated in the Kelly tumors (with high expression of MYCN oncogene), while a very high uptake was observed in the lungs. In contrast, in case of the intratumoral injection, the injected 191Pt-MYCN-PIP was retained in the Kelly tumors and not excreted (357 %IA/g at 4 d). Based on these results the authors concluded that the compound design should be improved to enable its delivery to a tumor via intravenous injection. However, these most valuable results of in-vivo studies are missing in the abstract and have to be added.   

In general, the manuscript I very well written; The study is well designed and based on the state-of-art methodologies. Due to the novelty of the design of the new radiotherapeutic agent for gene-targeted Auger electron therapy, the presented study is worth considering as the important contribution into the field. Some issues have to be addressed prior to publication.

-          Please add the data on the nuclear-physical properties of 191Pt such as decay route, the yield of AEs per decay, energy etc;

-          Lines 85-86: “191Pt-labeled MYCN/GCC-PIP was obtained in a high radiochemical yield of 50–70% (Fig. S2).” Fig. S2 represents HPLC chromatograms. It is not possible to evaluate the radiochemical yield from these data, one should know injected activity and product activity because very often the activity retains on the C18 column.  

-          It’s kind of surprising that PREP HPLC and analytical HPLC on Fig S2 looks so similar. Please provide the details for both separations – such as flow rates, injected volumes.

-          Please give the description of the radiolabeling procedure.

-          Lines 208-210. The sentence”This section may be divided by subheadings. It should provide a concise and precise description of the experimental results, their interpretation, as well as the experimental conclusions that can be drawn” looks like it was implemented from some other text, like “the reviewer’s evaluation”.

-          Please modify the abstract by adding the results of the in-vivo studies (see above). 

To conclude, the paper is recommended for publication in Pharmaceuticals after considering above questions and comments.

Reviewer 2 Report

Comments and Suggestions for Authors

Reviewer comments

The authors have presented their work synthesizing and, through in vitro and in vivo experiments, characterizing a radioplatinum-labeled pharmaceutical targeting the MYCN oncogene im neuroblastoma cancer models. The radionuclide payload of the drug is 191Pt, a unique and promising radionuclide emitting many low energy Auger electrons. The 191Pt is attached to the biological targeting vector through a single cysteine residue. Besides mass spec of the unlabeled vector, there was no chemical characterization of the radiolabeled conjugate or its non-radioactive standard. Additionally, the authors did not characterize the chemical stability of the radiopharmaceutical to de-platinization. The targeting vector utilizes a 12-ring pyrrole-imidazole polyamide (PIP) structure that is designed to target the MYCN gene, but efforts to demonstrate this binding specificity failed, evidenced by the fact that high and low MYCN expression level cells had identical cell and DNA uptake of the radiopharmaceutical (Fig 2c,2d). Despite these limitations, the manuscript contains several intriguing results that will advance the fields of Auger electron-  and PIP-based targeted therapeutics. I have included a list of my detailed suggestions to improve the manuscript below.

Radiopharmaceutical synthesis and chemistry

·      Were cold standards of natPt-MYCN-PIP and natPt-GCC-PIP synthesized and characterized by mass spec / HPLC? If not, why not? A crystal structure of the cold labeled compound would go a long way in helping understand the way the Pt is being bound by the vector.

·      Did the authors perform serum stability or cysteine challenge experiments for the 191Pt-MYCN-PIP conjugate? It’s very likely that free thiols or disulfides in blood proteins could outcompete with the single cysteine-Pt linkage that binds the radionuclide payload to the radiopharmaceutical.

·      It’s not clear who were the suppliers of the Fmoc functionalized monomer units

·      It’s not clear what the Fmoc-beta-alanine-OH was used for – there is no alanine in the structure reported in Figure 1

·      There must need to be a Fmoc-glycine-OH used as well, but not reported.

·      Line 297: How many equivalents of coumarin were added here? What volume of DMF?

·      Line 300: As this is a liquid phase conjugation step, it is unclear what is meant by ‘drained into Et2O”

·      Line 301: What volume of TFA/TIS/water was used for deprotection?

·      Line 302: Again it’s unclear what is meant by ‘drained into Et2O’ after this liquid-phase chemistry step

·      Line 304: what mobile phase composition/gradient was used for C18 preparative chromatography?

·      Line 320: The supplementary materials does not describe what the bulk solution composition for the 191Pt. It states the KCl and phosphate concentrations but is this in water?

·      Line 320: What was the MYCN/GCC-PIP dissolved in for addition to the radiolabeling reaction? Is the 2 mM referring to the 3µL solution added or the entire 230µL reaction mixture?

·      Line 326: amount of Tween 80 added?

·      The supplementary methods describes a high purity gamma spectrometer, but no where does it describe this being used.

Cell-binding assay:

·      For these cell experiments, the volume of media over the cells was not reported. This is important in estimating the concentration of the radiopharmaceutical and should be included.

·      Line 126/128/130/131: The concentration of non-radioactive MYCN-PIP or GCC-PIP incubated with the cells should be added to the figure description for parts c and d

·      Line 347/Figure 2b: What did the autoradiograph of 191Pt-MYCN-PIP without oligonucleotide look like?

·      Figure 2b: What would an autoradiograph of oligonucleotide + free 191Pt look like?

·      Figure 2b: What would an autoradiograph of oligonucleotide + 191Pt-GCC-PIP look like?

·      Why weren’t these controls run? They would be useful in confirming the stability of the 191Pt-MYCN bond and the specificity of the MYCN=PIP over the GCC-PIP.

·      Line 344: What was the purpose of including the 10mM sodium dimethylarsenate in the gel electrophoretic mobility shift assay? This does not seem to be a common reagent for this assay.

·      Line 369: Why was the non-radioactive MYCN-PIP or GCC-PIP concentration varied so significantly with these experiments. I would anticipate that this could have a major impact on the amount of 191Pt uptake in these experiments.

DNA-binding assay:

·      For these cell experiments, the volume of media over the cells was not reported. This is important in estimating the concentration of the radiopharmaceutical and should be included.

·      Line 126/128/130/131: The concentration of non-radioactive MYCN-PIP or GCC-PIP incubated with the cells should be added to the figure description for parts c and d.

·      Line 380: Did the authors confirm that the retention/elution of the 191Pt activity from the DNA extraction kit was not due to retention/elution of 191Pt or 191Pt-MYCN-PIP or 191PtGCC-PIP on the silica column? This could be confirmed by doing a processing a cell-free control through the DNA binding kit and ensuring no 191Pt activity is eluted in the kit in the elution buffer.

·      Line 380: Did the cells treated with 191Pt-GCC and the cells treated with 191Pt-MYCN have statistically different masses of DNA? In other words, was the significant different in %IA/mg gDNA between the two due to difference in 191Pt activity or due to differences in DNA mass?

53BP1-EGFP foci experiment:

·      For these cell experiments, the volume of media over the cells was not reported. This is important in estimating the concentration of the radiopharmaceutical and should be included.

·      Line 115: This statement is not supported by Fig S3. The micrographs must be quantified and statistics must be made comparing the 191PtMYCN-PIP treated cells with MYCN-PIP-treated cells with untreated cells.

·      Line 115: Was a control treatment of 191Pt-GCC-PIP treated cell made for comparison with the 191Pt-MYCN-PIP-treated, MYCN-PIP-treated and untreated cells?

Cytotoxicity assay

·      For these cell experiments, the volume of media over the cells was not reported. This is important in estimating the concentration of the radiopharmaceutical and should be included.

·      Line 137: The statements of results should be supported by statement of the measured values, standard deviation and number of replicates. Comparison of the results to one another should be supported by a statement of the significance.

·      Fig 3a should have significant differences noted using the ns,*,** nomenclature used in Fig 3b.

RT-qPCR experiments:

·      For these cell experiments, the volume of media over the cells was not reported. This is important in estimating the concentration of the radiopharmaceutical and should be included.

·      Figure 3b and comparison with the results section is very confusing. The legend does not state what the dotted bars represent. The results text implies that Figure 3b-left should be a comparison of the MCYN-expression levels of the three PBS-treated cell lines (in the absence of MCYN-PIP or 191Pt-MYCN-PIP), however, this does not agree with Figure 3b. It is not clear why the results presented in Figure 3b-right were not normalized to the respective cell lines’s MYCN-expression levels of the PBS treated cell lines, as opposed to the MYCN-PIP-treated cells. That way the reader could see if expression levels changed with treatment of the non-radioactive pharmaceutical (this is how the authors chose to present the IFN PCR results in Fig 4b).

·      The RT-qPCR methods say that an 18S rRNA control assay was used, but there was no description of how or if this was used to normalize the expression data.

·      In the RT-qPCR experiments, the cells were treated with the same activity concentration for longer time compared with the viability experiments. Therefore, we know that there will be fewer viable cells in the cells cultured with 191Pt-MYCN-PIP. What gives the authors confidence that the difference in expression levels observed in the PCR experiments were not simply a manifestation of there being fewer viable cells from which DNA was able to be extracted and amplified? Did this somehow get normalized for using the 18S rRNA control assay?

·      Nowhere in the results, Fig3 description or methods do the authors state in what number of replicates were the PCR experiments performed.

·      Figure 3 description does not explain the meaning of *, **, and ns.

FISH experiments:

·      For these cell experiments, the volume of media over the cells was not reported. This is important in estimating the concentration of the radiopharmaceutical and should be included.

·      Line 425-427: The authors repeat themselves in two successive sentences. I appreciate the authors including this important methodological detail, but it does not need to be included twice.

·      This FISH data is very convincing. Is there anything else that could be causing this other than a decrease in MYCN gene content in the cells? Like arresting the cycle in a specific cell cycle phase? If it was arrested at a part of the cell cycle where the DNA is more or less condensed, then maybe that would affect the detection capability of the FISH probe?

·      Was the FISH experiment not done with the 191Pt-GCC-PIP control?

Immunofluorescence:

·      The results section is written to imply that only cytosolic dsDNA was quantified in this assay. However, it’s clear from the methods that nuclear and cytosolic fluorescence was both quantified in Figure 4. I suggest changing line 176 to read, ‘… the whole cell fluorescence signal was…’

·      From the images in Fig 4, it does appear that there is more cytosolic red fluorescence, but even more strikingly there is MUCH more nuclear red fluorescence. Why is there so little red fluorescence in the nuclei of the control cells? This indicates to me that there may be a problem in the normalization of these images and intensity quantification.

·      In Figure 4, the meaning of ‘****’ is not explained in the figure description.

Biodistribution

·      How does the BioD compare to the 18F-labeled PIP images from references? Why might this be different?

·      How does the BioD compare with injected free 191Pt? If it’s similar, what might that indicate?

·      Line 208: remove the text, “This section may be divided by subheadings. It should provide a concise and precise description of the experimental results, their interpretation, as well as the experimental conclusions that can be drawn.”

·      I recommend putting the kidney and bone on the scale of the right panel of Figure S4b

·      If a full biodistribution following intratumoral injection was completed, I recommend reporting the full organ level biodistribution results.

·      There appears to be a disagreement between Fig S4b-right and S4c. In S4b-right, it appears as though the tumor uptake at 2 min is smaller than that of both the 1 day point. However, it appears to have greater uptake at 2 minutes in Fig S4c.

Discussion

·      Paragraph 1: This text should mention specific figures that support the statement. This was done for discussion paragraph 3, but not paragraph 1.

·      Paragraph 2, line 231: It also seems possible that the adjacent triargenines contributed to the binding to Pt through their nitrogen-rich guanidine moieties. 

·      Line 237-239: I also believe that the lack of radioactivity in the unshifted band indicates that 191Pt is not

·      Line 252: Needs re-writing, “these results expect…”

·      Paragraph 6, lines 274-282: This discussion of the biodistribution should be augmented with comparison with the literature descriptions of the biodistribution of free Pt and 18F-PIPs.  For example, from Harki et al PNAS 105(35):13039 (2009), I’d expect the PIP to be primarily biodistributed to the liver. Based on Obata et al Nuc Med Comm 43:1121 (2022), I’d expect cisplatin (nearly inorganic Pt) to go to lung and kidney.

Conclusion

·      I suggest changing the first sentence to be more precise than to simply say that ‘191Pt-MYCN-PIP works in living cells.’ This is overly broad and ambiguous

·      A statement about how the compound had identical cell/DNA uptake in MYCN+ and MYCN- cells should be added to conclusions.

·      A statement about this compound’s problems in vivo should be added to the conclusions

Additional minor comments:

·       In multiple places in the manuscript, the results section uses acronyms that are now explained until the methods and materials section (e.g. FISH, PBS, etc)

·       Line 87: Unexplained acronym: HPLC,

·       Line 76: Unclear what the superscripted ‘191Pt,11’ is meaning.

·       Line 121: Remove text, “This is a Figure. Schemes follow the same formatting”

Comments on the Quality of English Language

The quality of the English is largely good. There are several locations where it can be improved, as stated in the 'Comments and Suggestions for Authors' section above.

Round 2

Reviewer 1 Report

Comments and Suggestions for Authors

Dear authors, thank you very much for your detailed response. I am fully agree with all your comments and the manuscript text corrections. Congratulations with an excellent article.

Author Response

We greatly appreciate the encouraging comment.

Comment: Dear authors, thank you very much for your detailed response. I am fully agree with all your comments and the manuscript text corrections. Congratulations with an excellent article.

Our response: Thanks for the positive and fruitful review!

Reviewer 2 Report

Comments and Suggestions for Authors

Line 438: This sentence is worded confusingly. Perhaps it should read, “The radiochemical yield was evaluated as the radioactivity ratio of the purified product to the crude sample injected onto the preparative HPLC.”

Figure 2b, re: autoradiograph of oligonucleotide+free 191Pt: Why would 90 kBq of no carrier added 191Pt cross link 5µM oligonucleotide? For 5 µM oligonucleotide in 100 µL, this would be 500 pmol of oligonucleotide. If carrier-free, we’d expect 90 kBq to have 0.05 pmol of Pt. If you are indeed observing DNA cross linkage as a result of the addition of 90 kBq of free 191Pt, it seems much more likely that this cross linkage is as a result of the residual Ir present in the 191Pt preparation. Was the Ir mass (or a Ir/Pt separation factor of the TBPresin+AX separation process) in the final 191Pt solution ever measured?

Line 542: I suggest you change text to read, “…based on the comparative cycle threshold (CT) method (ΔΔCT method).”

Line 201, 589: In the figure descriptions and Statistical Analysis section, please change ‘ns ≥ 0.05’ to ‘ns: p≥0.05’.

Figure 3a: I am surprised the difference between the PBS-treated and 191Pt-MYCN-PIP-treated Kelly cells does not show significance. Please note the on the figure with the ‘ns’ nomenclature.

Figure 3b left: the legend still confuses me. I suggest just removing the legend. Also, in this part of the figure, the SK-N-AS relative expression looks to be zero. If a relative expression was measured, perhaps breaking the y-axis of the plot to be able to convey the SK-N-AS relative expression of MYCN.

Figure S4: If the biodistribution experiment was performed twice, perhaps you should collect the data into a single cohort and the average of all animals reported. Were there experimental differences between the two experiments that give you hesitancy to do this?

Thank you for your description of your experience with the free 191Pt biodistribution in mice. This seems like valuable insight to add to the manuscript. Is the data publishable? If not, is it possible to add it to the supplementary material of this manuscript? If not, perhaps it could just be cited as, ‘data not shown’. Or ‘data to be published in forthcoming manuscript’ or something like that.
